# Augmented Corticotomy on the Lingual Side in Mandibular Anterior Region Assisting Orthodontics in Protrusive Malocclusion: A Case Report

**DOI:** 10.3390/medicina58091181

**Published:** 2022-08-30

**Authors:** Yun Lu, Haohao Liu, Jialiang Liu, Meihua Chen

**Affiliations:** 1Department of Orthodontics, Shanghai Stomatological Hospital & School of Stomatology, Fudan University, Shanghai 200001, China; 2Department of Periodontology, Shanghai Stomatological Hospital & School of Stomatology, Fudan University, Shanghai 200001, China; 3Department of Oral and Maxillary Surgery, Shanghai Stomatological Hospital & School of Stomatology, Fudan University, Shanghai 200001, China

**Keywords:** lingual corticotomy, bone graft, protrusive malocclusion, fenestration and dehiscence, mandibular incisor retraction

## Abstract

Adequate alveolar bone volume is a prerequisite condition for successful orthodontic tooth movement and posttreatment stability. Mandibular anterior teeth are more likely to exhibit dehiscence and fenestration in adult patients, which make orthodontic treatment in adults challenging, especially when the amount of retraction of the anterior teeth is large. Herein, we report the treatment of augmented corticotomy only on the lingual side in the mandibular anterior region to increase the volume of soft and hard tissue assisting orthodontics in a Class I bialveolar protrusive malocclusion and propose management strategies of mandibular incisor retractions. A 22-year-old female with a chief complaint of protrusive mouth presented to the Department of Orthodontics for orthodontic treatment, diagnosed with Class I bialveolar protrusive. The orthodontic treatment plan involved the extraction of four premolars and extensive retraction of the anterior teeth using microimplant anchorage. In consideration of the fenestration and dehiscence in the mandibular anterior alveolar bone and the pattern of tooth movement, augmented corticotomy was performed on the lingual side combined with bone grafting. Clinical and radiographic evaluation after treatment revealed significant improvements in the facial profile and in periodontal phenotype. Augmented corticotomy assisting orthodontic treatment could be a promising treatment strategy for adult patients with alveolar protrusion to maintain periodontal health.

## 1. Introduction

Lip protrusion is a frequently presenting chief complaint from adult patients in an orthodontic outpatient setting. To resolve the protrusive profile, extraction of four premolars and maximum retraction of anterior teeth are usually required. The maintenance of periodontal health plays a vital role during adults’ orthodontic treatment. Hence, it is necessary for adults to assess periodontal risk prior to orthodontic treatment. Cone-beam computed tomography (CBCT) is suggested for periodontal risk assessment and orthodontic treatment planning [1].

Anatomical limitations of tooth movement are vital at the beginning of the treatment planning stage. Orthodontic tooth movement is limited by the alveolar bone housing, which is a cortical plate of the alveolar bone surrounding the tooth [2]. If the tooth moves beyond the limitation of the alveolar bone housing, periodontal supportive tissue gets compromised, resulting in a poor prognosis [3]. Thus, further orthodontic tooth movement can be achieved with surgical assistance. The concept of a corticotomy evolved to include alveolar ridge augmentation to resolve the concern of the potential violation of the periodontal bony tissue. Currently a common surgical procedure, augmented corticotomy assists orthodontic treatment by extending the range of tooth movement and improving posttreatment stability [4].

For bialveolar protrusion malocclusion, to reduce facial convexity, extensive retraction of the mandibular anterior teeth is required. However, large amounts of movement of lower anterior teeth may pose great risk of bone loss and gingival recession. Augmented corticotomy may be used during orthodontic treatment in cases of the root being out or in major proclination movements in the anterior region. Nevertheless, due to the poor visual fields and difficult operation on the lingual surgical area, limited numbers of cases treated with lingual corticotomy have been reported in the literature. This case report illustrates that corticotomy on the lingual side assisting orthodontics in protrusive malocclusion could be a promising treatment to reduce the risk of periodontal hard- and soft-tissue loss.

## 2. Case Report

### 2.1. Diagnosis and Etiology

A 22-year-old female with a chief complaint of protrusive mouth presented to the Department of Orthodontics for orthodontic treatment. The patient was systemically healthy, and her personal and family history were noncontributory. She had received periodontal supportive treatment regularly before.

The patient presented with a Class I malocclusion on a skeletal Class I base, with an average mandibular plane angle and increased overjet. She exhibited profile convexity, a protrusive lower lip, and mentalis strain on lip closure. The intraoral clinical examination showed right second molar buccal crossbite and dental midline deviation. She also had a thin scalloped biotype with translucent gingiva, narrow keratinized tissue, and highly scalloped gingival margin (Figure 1).

The panoramic radiograph showed the slight absorption of the alveolar ridge in the lower anterior region (Figure 2A). The cephalometric evaluation showed that most dental and skeletal parameters were inside the standards of normality, while a greater degree of lower lip prominence (LL-E) was observed (Figure 2B,C and Table 1). CBCT examination revealed a very thin alveolar plate and vertical alveolar bone loss on the buccal and lingual sides (Figure 2D).

### 2.2. Treatment Objectives

The treatment objectives were to (1) retract anterior teeth, (2) improve the convex profile, (3) align and level the dental arches, and (4) improve the gummy smile.

### 2.3. Treatment Alternatives

Two treatment options were considered for this patient. The first option was extraction of four first premolars and retraction of anterior teeth. However, CBCT examination showed an inadequate alveolar plate on lingual side, indicating that a little retraction of the lower anterior teeth could not improve the aesthetics of the convex profile. The second option was extraction of four first premolars, miniscrew-assisted extensive retraction of anterior teeth, and lingual augmented corticotomy in lower anterior teeth. The treatment options were discussed with the patient, and she preferred orthodontic treatment to improve her convex profile. Considering the aesthetics of the convex profile, the patient preferred the second treatment options. The main dental and profile treatment objectives for this patient were to improve the convex profile by retracting the anterior teeth. In such cases requiring significant retraction of incisors, careful consideration of alveolar bone width and the integrity of the attached supporting tissue is required.

### 2.4. Treatment Progress

Passive self-ligating brackets (Damon Q, Ormco Corp., Orange, CA, USA) were placed in both arches. The archwire sequences were 0.014′′NiTi, 0.018′′NiTi, and 0.019 × 0.025′′NiTi for initial alignment and leveling, and a 0.019 × 0.025SS archwire at the final stage of leveling. Because the maxillary and mandibular arches had nearly no crowding, the four first premolars were retained until completing alignment and leveling, in order to preserve the width of the alveolar ridge as much as possible. Miniscrews (diameter, 1.6 mm; length, 10 mm; Hubit, Daegu, South Korea) were placed between the first and second molars bilaterally in the maxillary arch for assisting upper-incisor retraction. In order to provide more overjet for the upper anterior teeth during space closure, the retraction of the mandibular incisors began 1 month earlier than the maxillary, immediately after the extraction of the premolars (Figure 3A,B). Augmented corticotomy was performed by an experienced periodontist late in the space-closure phase (Figure 3C). The surgery was performed under local anesthesia. A full-thickness mucoperiosteal flap was reflected lingually with papillary preservation incision in both arches, from the second premolar to the second premolar (Figure 4B). The lingual flap reflection extended at least 5 to 10 mm beyond the apices of the teeth. Corticotomy was performed by piezoelectric instrumentation (UltraSurgery; Woodpecker, Guizhou, China) between the roots of the teeth, with a depth of 3 mm in the interproximal cortical bone, starting from 2 to 3 mm below the alveolar crest and extending to 5 mm beyond the root apices (Figure 4C). Bone-derivative material (Heal-All Bone Repair Material, 0.5 g; Zhenghai Bio-Tech, Yantai, China) was grafted onto the lingual aspect of the decorticated anterior cortical bone, over dehiscence and fenestration (Figure 4D). A bioabsorbable collagen membrane (Heal-All Oral Cavity Repair Membrane, 15 mm × 20 mm; Zhenghai Bio-Tech, Yantai, China) was adapted to completely cover the graft area (Figure 4E). Primary closure was achieved by flap repositioning to the gingiva papilla with interrupted interdental sutures (4-0 vicryl, polyglactin 910, 3/8 reverse cutting, Ethicon, Somerville, USA) (Figure 4F).

Postoperative instructions were reviewed (cefradine 500 mg, TID; Peridex 0.12%, 15 cc BID). Following surgery, sutures were removed 3 weeks later. Space closure continued in 2-week intervals and finished within 3 months after surgery (Figure 3D). The total orthodontic treatment period was 24 months. After removing the miniscrews and appliances, a vacuum-formed retainer was required for retention in both the maxilla and mandible. Supportive periodontal treatment was performed every 3 months during the orthodontic treatment.

### 2.5. Treatment Results

Following the treatment, good preservation of the interdental papillae and little gingival recession was observed, and gingival thickness was enhanced significantly (Figure 5). In addition, as the patient’s chief complaint was addressed, the treatment aims were accomplished. Good occlusal and aesthetic results were achieved (Figure 6). Final cephalometric analysis showed that SNB was increased by 0.8° and ANB reduced by 0.9°. Both MP-SN decrease of 1.1° and MP-FH decrease of 1.0° indicated a counterclockwise rotation of the mandibular plane. The upper incisor to the SN plane angle was slightly increased by 2.6° in the retraction of the upper labial segment. The lower incisor–mandibular plane angle was decreased by 14.5° to 90.7° suggesting retroclination of the lower labial segment with treatment. Aesthetically, the lower-lip position relative to Ricketts’s E plane was reduced by 4.7 mm, indicating improvement in her lateral facial profile, which was significantly due to the lower incisors’ retraction and uprighting and counterclockwise rotation of the mandible (Figure 7A and Table 1).

The cephalometric superimposition showed that the mandible rotated during orthodontic treatment in a tendency to decrease the mandibular plane angle, which would bring the chin upward and forward (Figure 7C). The changes that contributed to this tendency involved intrusion of upper incisors and molars, intrusion of lower incisors, and uprighting of lower molars. The increase in chin prominence results from a combination of forward rotation of the chin and retraction above the chin that alters the bony contours.

At the 1-year follow-up visit, CBCT examination showed the alveolar bone width had gained a lot at the different levels (Figure 7D). As shown in Table 2, on the bone-graft side, the lingual alveolar bone area had increased by 1.54 ± 0.33 mm^2^. As for the other side, the labial alveolar bone area had slightly increased by 1.04 ± 0.56 mm^2^. Radiographically, the mean alveolar bone thickness gain was between 0.72 ± 0.77 mm and 1.25 ± 0.67 mm at the different levels (Table 3).

The patient was satisfied with the final excellent treatment results and the improvement in the convex profile. In addition, she had no complaints of discomfort for periodontal surgery.

## 3. Discussion

The patient’s lateral facial profile was convex with mentalis strain on lip closure and a deficient chin, while a prominent bony chin was observed in the lateral cephalometric radiograph. The contour of the soft-tissue chin was affected by the shape of the bony chin and the area just above the chin, lower incisors, and corresponding alveolar process. In this case, the lower incisors needed maximum retraction and uprighting to achieve the desired chin contour. There were alveolar bone defects on the labial and lingual side of lower anterior teeth. It would be dangerous for lower incisor retraction, with a risk of loosening teeth and gingival recession. Therefore, augmented corticotomy was proposed as an adjunct to orthodontic treatment in this case.

Initially, the concept of corticotomy assisting orthodontic treatment stems from the regional acceleratory phenomenon [5]. However, only accelerating tooth movement, corticotomy cannot resolve the concern of the potential violation of periodontal bony tissue. Therefore, the alveolar corticotomy technique was modified by Wilcko, consisting of a selective partial decortication of the cortical plates and concomitant bone grafting and augmentation [6]. The replacement of bone-grafting materials increases bone thickness and density [7]. Corticotomy combined with bone graft can augment the alveolar bone, which makes tooth movement safe and effective. The added alveolar augmentation step is indicated in orthodontic situations where there is a concern that extensive tooth movement may move teeth out of the bony housing, resulting in bony defects, such as fenestration and dehiscence. Nowadays, augmented corticotomy has been proposed to improve periodontal phenotype to prevent negative outcomes [8].

In addition, different nonsurgical approaches, categorized into biological, physical, and biomechanical, have been created in order to accelerate orthodontic tooth movement [9]. Application of intermittent resonance vibrations, drug injections of vitamin D, prostaglandins, osteocalcin around the alveolar sockets, and low-level laser-light therapy (LLLT), called photobiomodulation therapy, can achieve rapid tooth movement and reduce treatment duration. Due to the presence of inadequate alveolar bone in this case, there were still a lot of uncertainties and unanswered questions regarding most of these techniques. Augmented corticotomy is a promising tooth-acceleration technique because of its various advantages in periodontal, aesthetic, and orthodontic aspects.

Proper risk assessment and diagnosis is the key to successful management of these cases in adults. Adequate alveolar bone volume is a prerequisite for successful orthodontic tooth movement and posttreatment stability [10]. Jing et al. found that mandibular anterior teeth were more likely to exhibit dehiscence and fenestration in adult patients [11]. These considerations make adults’ orthodontic treatment challenging, especially when the amount of retraction of the anterior teeth is large. The pattern of tooth movement in an orthodontic treatment plan can be another key factor in whether or not and where to perform periodontal surgery initially. Figure 8 shows common tooth-movement patterns of lower-incisor retraction. Understanding different tooth-movement patterns might help in the proposal of different treatment strategies. Figure 9 illustrates a proposed decision tree for the management of mandibular incisor-retraction cases. Specifically, the surgical indications are determined by the areas of alveolar bone defects and the range of retraction of anterior teeth.

In this case, pretreatment CBCT revealed the presence of dehiscence and fenestration on both the labial and lingual sides in the lower anterior region. Orthodontic treatment involved the controlled tipping movement of anterior teeth, and the alveolar thickness on the labial side of the teeth affected tooth movement little. Therefore, it was decided to perform augmented corticotomy only on the lingual side in this case. Generally, though, it is more difficult for a periodontist to perform corticotomy on the lingual side than on the labial side. Limited exposure and unclear view of surgery field on the lingual side is the obstacle for vertical decortication cuts, increasing corticotomy technique difficulty. In addition, due to rich and variable vascular supply in the floor of the mouth, some tricky complications may occur during corticotomy surgery [12]. Thus, thorough knowledge of the anatomical features of a surgery region is essential to avoid potential surgical complications. In addition, lingual tooth movement can result in increased thickness of hard and soft tissue at the labial aspect of the tooth [13]. In this case, gingival augmentation for prevention or treatment for gingival recession may not be needed. Proper application of the surgical approach with less invasion is necessary to achieve better outcomes.

It is believed that corticotomy stemming from initiating a regional acceleratory phenomenon (RAP) results in more rapid tooth movement [5]. This is consistent with RAP, which usually lasts for 1–2 months [14,15]. In this case, there was a 2 mm space in the mandibular arch before surgery, and 6 weeks later the space had closed. In addition, RAP could explain the increased rate of space closure in recent extraction sites. One suggested method to increase efficiency of space closure is to move teeth into a freshly extracted socket [16]. Additionally, the less calcified alveolar bone surrounding the extraction socket compared to that of the healed extraction site may have been another reason for the increased rate of space closure [17,18]. In this case, four first premolars were extracted after completing alignment and leveling of the maxillary and mandibular arches, and space closure began immediately after the extraction. We found that extraction of premolars immediately before space closure significantly increased the rate of space closure within the next 3 months.

Lingual augmented corticotomy assisting orthodontics was a good treatment option in the present case. Following orthodontic treatment combined with lingual augmented corticotomy, the patient was satisfied with the aesthetic outcome and periodontal health was under control. Both soft and hard tissue was stable clinically and radiographically. Corticotomy bone grafting increased the volume of alveolar bone, regenerated bone affected by dehiscence and fenestration, and avoided gingival recession. In addition, corticotomy and augmented bone grafting improved the thickness of the gingiva, which is particularly important to the patient with thin biotype. Some studies have reported bone formation and augmentation assessed by CBCT following surgery [19,20,21]. The increase in the volume of soft and hard tissue will help to improve the stability of the results of orthodontic treatment [22]. Several reports have examined the adverse effects to the periodontium after corticotomy, which ranged from no problems to slight interdental bone loss and loss of attached gingiva [23,24]. However, no such changes after treatment were observed in this case. The extra expenses for the surgery and the postoperative discomfort were the disadvantages of this technique that the patient needed to tolerate. Within the limitations of this case report, further assessment of consistency and stability of the outcomes over longer periods of time should be considered.

## 4. Conclusions

This single case demonstrated that augmented corticotomy via bone grafting is an effective treatment strategy to assist orthodontics for adults with lip protrusion to achieve desired facial aesthetics. Despite the weak nature of the evidence, in cases involving controlled tipping movement of retraction, corticotomy performed only on the lingual side can maintain periodontal health and increase the volume of soft and hard tissue. Proper application of the surgical approach with less invasion may achieve satisfactory outcomes.

## Figures and Tables

**Figure 1 medicina-58-01181-f001:**
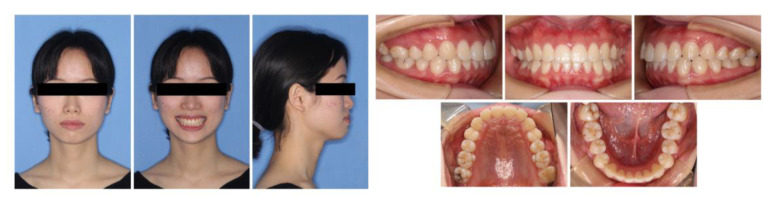
Pretreatment extraoral and intraoral photographs.

**Figure 2 medicina-58-01181-f002:**
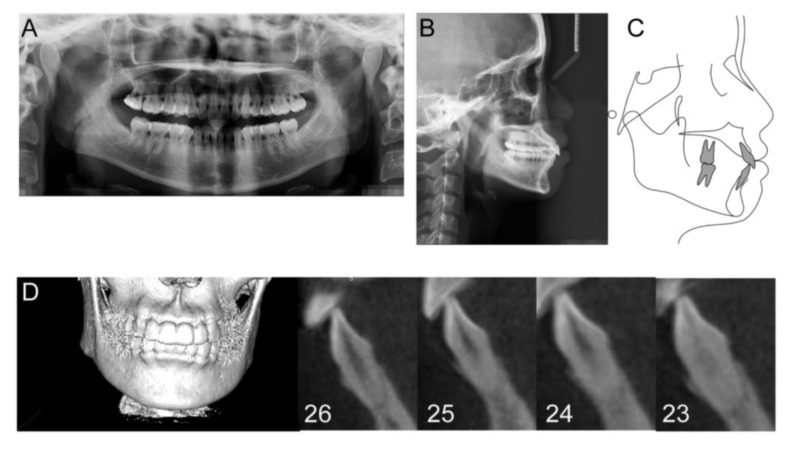
Pretreatment radiographs and cephalometric graph. (**A**) Panoramic radiograph; (**B**) Lateral cephalogram; (**C**) cephalometric tracings; (**D**) CBCT graphs.

**Figure 3 medicina-58-01181-f003:**
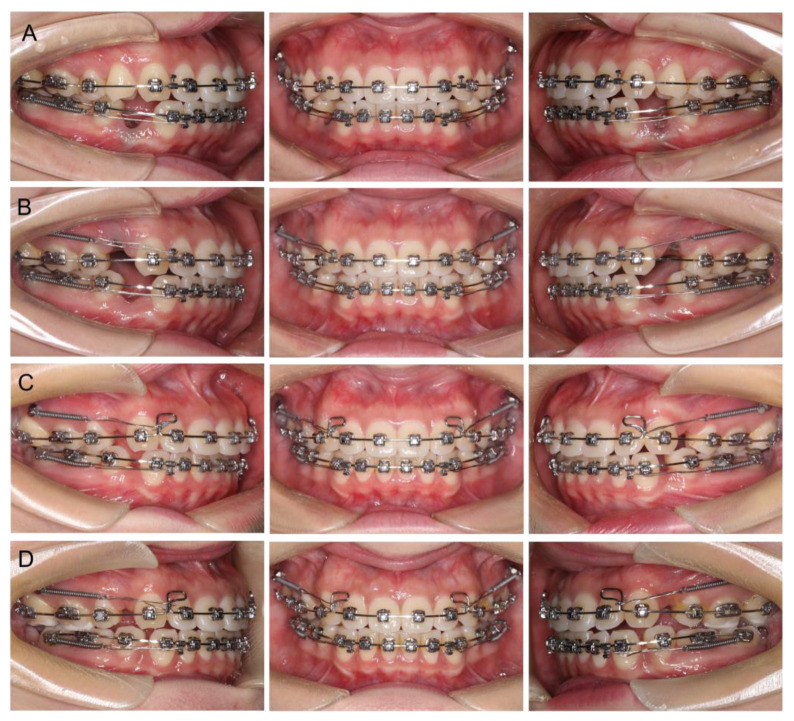
Clinical images of detailed treatment progress. (**A**) Retraction of the mandibular incisors immediately after the extraction of the premolars; (**B**) retraction of the maxillary incisors immediately after the extraction of the premolars; (**C**) before surgery; (**D**) space closure continued after surgery.

**Figure 4 medicina-58-01181-f004:**
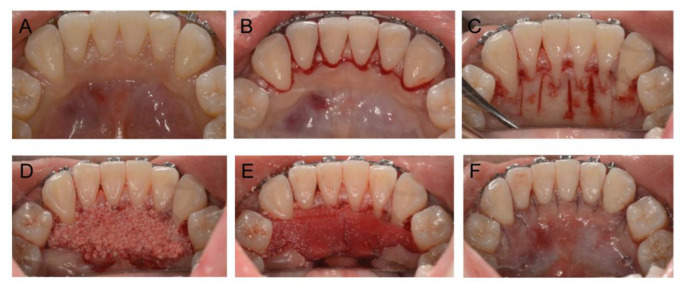
Clinical images of augmented corticotomy of the lingual side. (**A**) Initial intraoral image; (**B**) papillary preservation incisions; (**C**) piezoelectric corticotomy; (**D**) placement of bone grafts; (**E**) placement of bioabsorbable collagen membrane; (**F**) sutures.

**Figure 5 medicina-58-01181-f005:**
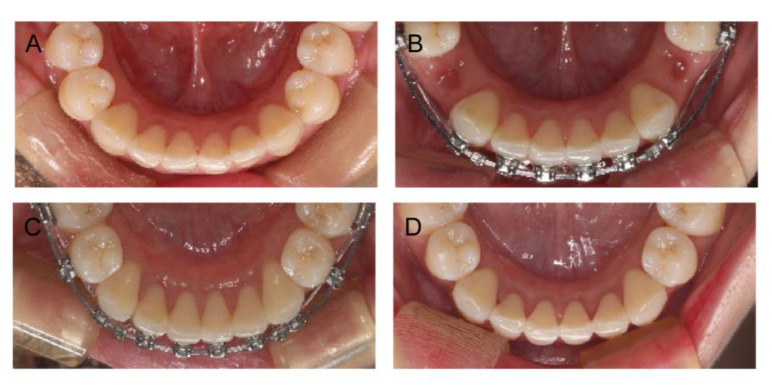
Pretreatment and posttreatment mandibular anterior teeth on lingual side. (**A**) Initial visit; (**B**) extraction of mandibular first premolars; (**C**) after augmented corticotomy; (**D**) orthodontic treatment completed.

**Figure 6 medicina-58-01181-f006:**
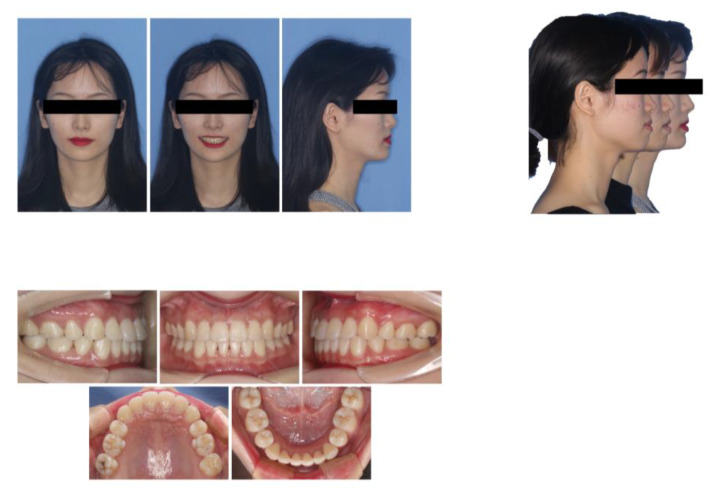
Posttreatment extraoral and intraoral photographs and superimposition photographs.

**Figure 7 medicina-58-01181-f007:**
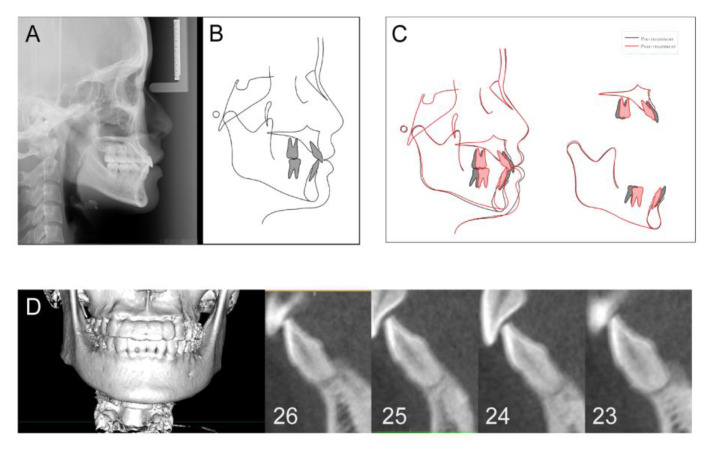
Posttreatment cephalometric graphs and CBCT graphs. (**A**) Lateral cephalogram; (**B**) cephalometric tracings; (**C**) cephalometric superimposition; (**D**) CBCT graphs.

**Figure 8 medicina-58-01181-f008:**
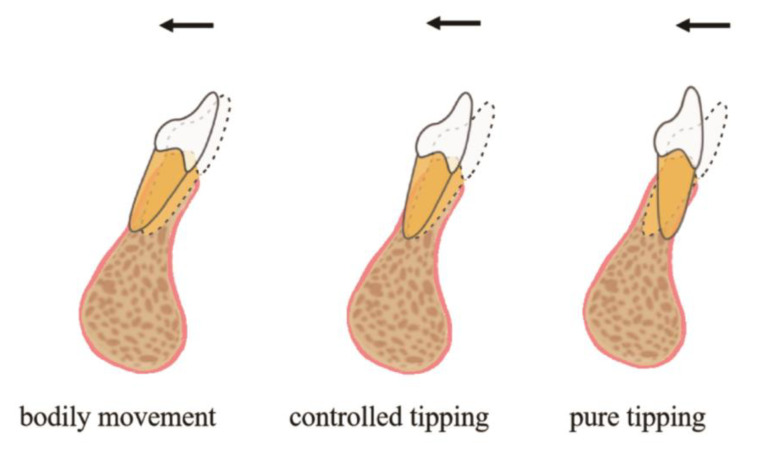
Tooth-movement patterns of retraction.

**Figure 9 medicina-58-01181-f009:**
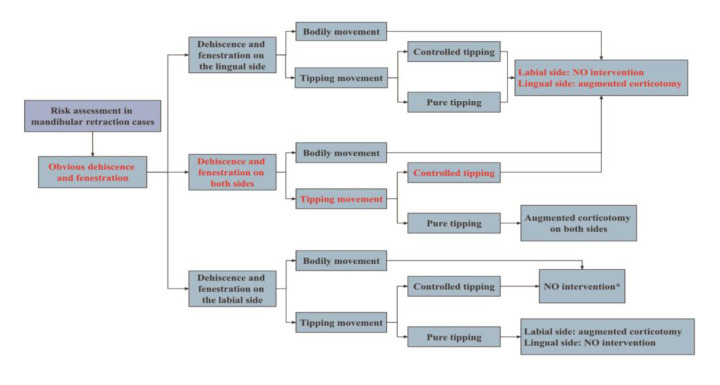
Decision tree for the management of mandibular incisor-retraction cases. * Extensive retraction may move teeth out of the bony housing, resulting in lingual bony defects, in which augmented corticotomy may be required despite no dehiscence and fenestration on the lingual side.

**Table 1 medicina-58-01181-t001:** Summary of cephalometric measurements.

Measurement	Normal (Standard Deviation)	Pretreatment	Posttreatment	Difference
SNA (°)	83.1 (2.7)	82.0	81.9	−0.1
SNB (°)	80.3 (2.6)	79.2	80.0	0.8
ANB (°)	2.7 (1.8)	2.8	1.9	−0.9
UI—SN (°)	103.4 (5.5)	105.8	108.4	2.6
LI—MP (°)	96.3 (5.4)	105.2	90.7	−14.5
UI-LI (°)	129.1 (7.1)	118.1	131.1	13
MP—SN (°)	32.6 (6.9)	30.9	29.8	−1.1
MP—FH (°)	25.5 (4.8)	26.1	25.1	−1.0
Wits (mm)	−1 (1)	−0.8	−1	0.2
PP-OP (°)	10 (4)	7.3	5.1	−2.2
P-A Face Height (%)	65 (4)	67.5	68.4	0.9
Y-Axis (°)	67 (5.5)	68.2°	66.7	−1.5
UL-E (mm)	−1.6 (1.5)	−0.1	−1.8	−1.7
LL-E (mm)	−0.2 (1.9)	4.1	−0.6	−4.7

SN, sella-nasion plane; MP, mandibular plane; FH, Frankfort horizontal plane; OP, occlusal plane; PP, palatal plane.

**Table 2 medicina-58-01181-t002:** Alveolar bone area changes surrounding mandibular anterior teeth one year postoperation.

Tooth	26	25	24	23	Mean ± SD
Lingual (mm^2^)	1.49	1.74	1.82	1.09	1.54 ± 0.33
Labial (mm^2^)	0.68	1.86	0.68	0.94	1.04 ± 0.56

**Table 3 medicina-58-01181-t003:** Mean of lingual bone thickness gain one year postoperation.

Tooth	26	25	24	23	Mean ± SD
Coronal (mm)	0.40	0.63	0.30	2.11	0.86 ± 0.84
Middle (mm)	1.14	0.88	0.76	2.22	1.25 ± 0.67
Apical (mm)	0.26	1.14	0.31	1.87	0.72 ± 0.77

## Data Availability

All experimental data to support the findings of this study are available by contacting the corresponding author.

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
