# Peer review of "Augmented Corticotomy on the Lingual Side in Mandibular Anterior Region Assisting Orthodontics in Protrusive Malocclusion: A Case Report"

_medicina, 2022, doi:10.3390/medicina58091181_

Round 1
Reviewer 1 Report
Dear Authors,
thank for your interesting paper full of sparks for readers.
Only one observation:
Line 35 Lip protrusion is a frequent presenting chief complaint from adult patients in orthodontic outpatient setting. ... not so frequent in this moment when patients love protrusive lips
And
Please explain the orthodontic technique (the mechanic) that you used in the lower arch after the surgery
Author Response
Response to Reviewer 1 Comments
Point 1: Line 35 Lip protrusion is a frequent presenting chief complaint from adult patients in orthodontic outpatient setting. ... not so frequent in this moment when patients love protrusive lips.
Response 1: Thanks a lot for your suggestion. Line 35, the statements of “Lip protrusion is a frequent presenting chief complaint from adult patients in orthodontic outpatient setting” was corrected as “Lip protrusion is one of frequent presenting chief complaints from adult patients in orthodontic outpatient setting.”
Point 2: Please explain the orthodontic technique (the mechanic) that you used in the lower arch after the surgery.
Response 2: Thank you for your reminder. Line 145-146 “Space closure continued in 2-week intervals and finished within 3 months after surgery (Figure 3D).” was pointed to explain the orthodontic technique after surgery.
Reviewer 2 Report
This article is only the presentation of a case.
The result achieved in one case cannot be generalized.
Moreover we do not have at least a similar orthodontic case differently treated to be used as a control.
Despite the good clinical result achieved in this case , the conclusions of the article cannot be generalized.
The article is a simple case report.
Author Response
Response to Reviewer 2 Comments
Point 1: This article is only the presentation of a case. The result achieved in one case cannot be generalized. Moreover we do not have at least a similar orthodontic case differently treated to be used as a control.
Response 1: We would like to express our sincere thanks for your insightful and constructive comments. Actually this article is one single presentation of our case series. We have also performed corticomy in similar cases which are still under treatment so far. Later, case-control studies will be strictly designed to dig into the reasons for outcomes. Based on good clinical results achieved in the article, we propose a decision tree for the management of mandibular incisor retraction cases for the readers to refer to.
Point 2: Despite the good clinical result achieved in this case , the conclusions of the article cannot be generalized. The article is a simple case report..
Response 2: Special thanks to you for your good comments. It is really hard to draw any conclusion with powerful evidence from just one single case. We should be more cautious in drawing the conclusion and made correction in this section (Line 292-299).
Reviewer 3 Report
Dear authors,
Your manuscript presents an interesting method of treatment.
However, I believe that, due to the fact that you have focused too much on the description of the procedure itself, You have omitted a broader presentation of the case.
First of all, please divide the paragraph in following subsections
1. Treatment plan
a) Please provide treatment plan in points and please provide information why such a way of treatment was chosen
2. Alternative treatment
You provided Yourself within the article information that:” The treatment options were discussed with the patient and informed consent was obtained”. Please provide information what kind of alternative treatment was proposed and it was rejected.
3. Treatment progress
It is described quite fine, but please provide a photos (at least 2 additional sets) of patient. (e.g. after TAD insertion and at follow-up visit).
Please describe the patient's feelings during follow-up and the stability of treatment results.
Please discuss non-surgical methods supporting the movement of teeth (e.g. LLLT, photobiomodulation) and please discuss its usefulness in such a case.
Author Response
Response to Reviewer 3 Comments
Point 1: However, I believe that, due to the fact that you have focused too much on the description of the procedure itself, You have omitted a broader presentation of the case.
First of all, please divide the paragraph in following subsections
Response 1: We would like to express our sincere thanks for your insightful and constructive comments. We have divided this paragraph into five subsections according to the your suggestion (Line 63-196): Diagnosis and etilogy; Treatment objectives; Treatment allternatives; Treatment progress and Treatment results.
Point 2: Please provide treatment plan in points and please provide information why such a way of treatment was chosen
Response 2: Thanks a lot for your suggestion. We have provided treatment objectives in points and explained why this treatment opinion was chosen in next subsections (Line 92-104).
Point 3: You provided Yourself within the article information that:” The treatment options were discussed with the patient and informed consent was obtained”. Please provide information what kind of alternative treatment was proposed and it was rejected.
Response 3: Thank you for your valuable comment. We have provided treatment alternatives and detailed why the patient chose such a way of treatment (Line 92-104).
Point 4: It is described quite fine, but please provide a photos (at least 2 additional sets) of patient. (e.g. after TAD insertion and at follow-up visit).
Response 4: Special thanks to you for your good comments. 4 sets of clinical images of detailed treatment progress were added in Figure 3.
Round 2
Reviewer 3 Report
The authors have significantly improved the manuscript. I think that it can be accepted for publication. I would like only to ask the authors to provide CARE checklist in form of supplementary material. After adding this checklist, in my eyes, the manuscript will meet all the required standards. CARE Checklist — CARE Case Report Guidelines (care-statement.org)
Author Response
Point 1: The authors have significantly improved the manuscript. I think that it can be accepted for publication. I would like only to ask the authors to provide CARE checklist in form of supplementary material. After adding this checklist, in my eyes, the manuscript will meet all the required standards.
Response 1: We would like to express our sincere thanks for your constructive suggestion. We have provided CARE checklist in form of supplementary materials. Please see the the attachment.
